# Quantitative Analysis of Intracranial Atherosclerosis and Its Correlation with Ischemic Cerebrovascular Disease and Prognosis

**DOI:** 10.3390/brainsci15091009

**Published:** 2025-09-18

**Authors:** Jingjing Cai, Sizhan Chen, Shiyu Hu, Lijie Ren, Gelin Xu

**Affiliations:** 1Department of Neurology, The First School of Clinical Medicine, Southern Medical University, Guangzhou 510515, China; caijingjing@szseyy.com; 2Department of Neurology, Shenzhen Second People’s Hospital, The First Affiliated Hospital of Shenzhen University, Shenzhen 518035, China; chensizhan@szseyy.com (S.C.); hushiyu@szseyy.com (S.H.)

**Keywords:** intracranial atherosclerosis, vulnerable plaque, ischemic stroke, high-resolution magnetic resonance vessel wall imaging, plaque characteristics, quantitative analysis

## Abstract

***Background:*** Intracranial atherosclerosis disease (ICAD) represents a significant etiology of stroke. This study aimed to evaluate correlations between intracranial atherosclerotic burden and risk of ischemic events. ***Methods***: In this prospective observational study, all enrolled patients underwent High-Resolution Magnetic Resonance vessel wall Imaging (HR MR-VMI) within two weeks of onset, or of enrollment. Baseline assessments included modified American Heart Association plaque type, stenosis degree, intra-plaque hemorrhage (IPH), plaque thickness, plaque length, and vessel wall enhancement. Modified Rankin Scale (mRS) was followed with one-year treatment in adherence to the guidelines. Comparative analyses were conducted between symptomatic and asymptomatic groups, culprit versus non-culprit plaques, and favorable versus poor prognosis groups. ***Results:*** The study included 129 symptomatic and 42 asymptomatic patients. Hypertension, diabetes, and smoking were more prevalent in patients in the symptomatic group. Vulnerable plaque (97.7% vs. 64.3%, *p* = 0.003), IPH (17.8% vs. 4.8%, *p* = 0.022) and higher stenosis degree (χ^2^ = 2.675, *p* = 0.008) were significantly more prevalent in the symptomatic group. Culprit plaques were predominantly located in the superior wall of the middle cerebral artery (MCA) (χ^2^ = 15.561, *p* = 0.001) and the left wall of the basilar artery (χ^2^ = 34.138, *p* = 0.008). Factors associated with poor prognosis included older age (63.63 ± 8.19 vs. 55.63 ± 13.15, *p* = 0.001), presence of IPH (31.82% vs. 14.29%, *p* = 0.037), and elevated D-dimer levels (0.77 ± 0.60 vs. 0.40 ± 0.36, *p* = 0.022). ***Conclusions:*** Vulnerable plaque, specific lesion locations, and higher stenosis degree are significantly associated with ischemic events in ICAD. While plaque enhancement and stenosis correlate with stroke occurrence, they show no clear association with prognosis. Neither the length nor the thickness of plaques manifests a significant correlation with either stroke events or the prognostic outcomes.

## 1. Introduction

Intracranial atherosclerosis is a more common cause of ischemic stroke than extracranial atherosclerosis in Asian populations [1]. Intracranial atherosclerosis represent 30–50% of stroke etiology in Asian people [2]. Previous studies suggested that intracranial atherosclerosis disease (ICAD)-caused stroke was associated with high risk of recurrence and mortality [3]. Therefore, identifying characteristics of intracranial atherosclerosis is crucial for improving stroke outcomes.

Current management of ICAD primarily involves pharmacotherapy—including statins and antithrombotics—as well as endovascular interventions [4,5]. Treatment decisions are largely guided by the degree of arterial stenosis. Despite these interventions, the recurrence rate of ischemic stroke is as high as 20% [6]. Growing evidence from recent systematic reviews and meta-analyses underscores a critical limitation of this approach: a significant proportion of ischemic events originate from arteries with non-severe stenosis (<50%), suggesting that factors beyond lumen narrowing are key drivers of symptomatology [7,8].

This paradigm shift has been fueled by advances in neuroimaging, particularly high-resolution magnetic resonance vessel wall imaging (HR MR-VWI), which enables the direct visualization and detailed quantification of atherosclerotic plaques, including their size, morphology, and composition [9,10]. HR MR-VWI has emerged as the first-choice modality for identifying intracranial vulnerable plaques, with features such as positive remodeling enlargement of the outer vessel wall), intra-plaque hemorrhage (IPH), and enhanced plaque enhancement (indicating active inflammation and neovascularization) [11,12]. It has been proved to have high consistency with pathological findings [13,14].

Evidence-based medical evidence has confirmed the link between vulnerable plaques and ischemic stroke. A comprehensive meta-analysis of 33 clinical studies has shown that there is a strong association between vulnerable plaques on HR MR-VWI and an increased risk of future stroke [15]. Previous studies have confirmed that HR MRI-VMI can be used to evaluate the efficacy of statins in treating symptomatic intracranial atherosclerotic plaques [10]. This positions plaque imaging not just as a diagnostic tool but as a potential cornerstone for personalizing risk assessment and tailoring more aggressive therapeutic strategies for high-risk individuals. HR MR-VMI helps to identify high-risk population subgroups to test interventions that might reduce the risk of stroke recurrence [16].

This study utilized HR MRI-VMI to systematically quantify the morphological characteristics, composition, and distribution of intracranial arterial plaques. Correlations between vulnerable plaque features and the occurrence of ischemic stroke were revealed, along with their potential impact on patient prognosis. The novelty of this work lies in the proposal of a non-invasive and reproducible quantitative plaque analysis method, which offers significant clinical utility by providing reliable imaging biomarkers. This approach facilitates early intervention in stroke and contributes to improved patient prognosis, thereby supporting clinical decision-making for patients.

The remainder of this paper is organized as follows. Section 2 introduces the methods. Section 3 presents the results. Finally, Section 4 provides a discussion of the findings, draws conclusions, and suggests directions for future work.

## 2. Methods

### 2.1. Patients

This prospective observational cohort study enrolled patients continuously who were hospitalized in the neurology department from January 2021 to December 2023. The Shenzhen Second People Hospital’s Institutional Review Board approved the study protocol (approval number: KY2024-180-01PJ), and informed consent was obtained from all participants. All patients underwent a thorough workup including CT angiography (CTA), carotid ultrasound, cardiac ultrasound, Transcranial Doppler (TCD) with bubble injection, 24 h holter electrocardiogram, and laboratory tests for coagulopathies, infections, and occult malignancies. Patients over 18 years of age with ischemic stroke within 7 days of their first onset and confirmed by MR Examination to have a new infarction were included. Individuals with the following conditions were excluded: (1) cardiogenic strokes; (2) over 50% stenosis of the extra-cranial artery; (3) vasculopathy other than atherosclerosis, such as vasculitis, moyamoya disease and dissection; (4) contraindications to MR imaging or unsatisfactory image quality of HR MR-VWI; (6) history of stroke or transient ischemic attack; (7) intracranial artery occlusion; and (8) refusal.

Based on the study objectives, participants were categorized into different groups. The symptomatic group consisted of individuals who experienced acute cerebral infarction or transient ischemic attack within two weeks of onset, whereas the asymptomatic group included individuals without ischemic cerebrovascular events identified during physical examination. The favorable prognosis group was defined as patients with a modified Rankin Scale (mRS) score of less than 3, and 3 to 6 were divided into the poor prognosis group.

### 2.2. Baseline Assessment

Data collected included demographics, risk factor, imaging findings, treatments given, stroke severity, final clinical diagnosis, the Trial of ORG 10172 in Acute Stroke Treatment (TOAST) classification for classifying stroke [17], and outcomes. The vascular risk factors included hypertension (yes or no), hyperlipidemia (yes or no, including low-density lipoprotein cholesterol, high-density lipoprotein cholesterol, and triglyceride levels, measured in mmol/L), diabetes (yes or no), renal insufficiency (yes or no, with glomerular filtration rate < 60 mL/min/1.73 mm^2^), D-dimer, high sensitivity C-reactive protein, smoking status (never smoking, current or former), and family history of early onset of stroke (yes or no, first-degree relatives of men < 55 years old or women < 65-year-old have a stroke). In addition, the results of cervical vascular examination were also collected. All patients in this study underwent MRI examination within two weeks after stroke onset, using a Siemens 3.0T magnetic resonance instrument equipped with a dedicated head coil. The examination included axial T1-weighted imaging (T1WI), T2-weighted imaging (T2WI), T2-fluid attenuated inversion recovery sequence (FLAIR), diffusion-weighted imaging (DWI), time-of-flight magnetic resonance angiography (TOF MRA), plain scan, and enhanced black-blood T1WI. Gadopentetate dimeglumine was used as the contrast agent for enhancement.

The scan parameters were set as follows: For T1WI, the repetition time (TR) is 2000 ms and the echo time (TE) is 7.5 ms, with a slice thickness of 6 mm and a slice spacing of 1.8 mm. For T2WI, TR is 4000 ms and TE is 117 ms, also with a slice thickness of 6 mm and a slice spacing of 1.8 mm. For FLAIR, TR is 9000 ms and TE is 81 ms, having a slice thickness of 6 mm and a slice spacing of 1.8 mm. For DWI, TR is 3000 ms and TE is 55 ms, with a slice thickness of 6 mm and a slice spacing of 1.8 mm. For MRA, TR is 21 ms and TE is 3.43 ms, with a slice thickness of 0.6 mm and a voxel size of 0.6 mm × 0.6 mm × 0.6 mm. For the 3D-T1 SPACE sequence, TR is 750 ms and TE is 24 ms, with a slice thickness of 0.35 mm, a matrix of 408 × 512, and a voxel size of 0.6 mm × 0.6 mm × 0.6 mm.

Image evaluation was performed independently by two trained neuro-radiologists using medical imaging viewer software (Siemens Medical Systems, MAGNETOM Prisma 3.0T), both blinded to clinical information and each other’s assessments. If the result error range is greater than 20%, a third physician with 10 years of experience will be invited to participate in the decision. Based on the overall signal-to-noise ratio and image quality, those who could not complete the plaque analysis were excluded from this study. The clinical symptoms of the included ischemic stroke patients were consistent with the culprit vascular supply area, so as to ensure that atherosclerotic plaque was the responsible vessel for ischemic events as much as possible.

Plaque characteristics include (1) location: the basilar artery and the intracranial segment of the internal carotid arteries, M1 segments of middle cerebral artery, A1 segments of anterior cerebral artery, P1 segments of posterior cerebral artery, and intracranial segments of the vertebral artery; (2) maximum wall thickness; (3) maximum plaque length; (4) grade of plaque contrast enhancement; (5) luminal stenosis; (6) intra-plaque hemorrhage (IPH); and (7) plaque typing.

Maximum plaque length was defined as the longitudinal extent of the plaque along the vessel’s long axis. The level of luminal stenosis in intracranial arteries was assessed using maximum intensity projection reconstruction on TOF MRA images, following the criteria established by the North American Symptomatic Carotid Endarterectomy Trials (NASCT standards) [18]. The enhancement of atherosclerotic plaques was classified into three grades on post-contrast T1WI images by using published criteria [19]: Grade 1 indicates no enhancement, signifying a stable plaque without significant inflammation (Figure 1A); Grade 2 reflects mild-to-moderate enhancement, suggesting some inflammatory activity within the plaque (Figure 1B); and Grade 3 denotes high enhancement, which is associated with active inflammation (Figure 1C). Intracranial atherosclerotic plaques were classified into seven types (I to VII) according to the modified American Heart Association (AHA) classification criteria [20]. Types IV, V, and VI were designated as unstable plaques, also referred to as vulnerable plaques, while types I, II, III, VII, and VIII were categorized as stable.

Furthermore, HR MR-VMI was also used to determine the spatial distribution of plaques. For the middle cerebral artery (MCA), plaque locations were identified as superior, inferior, anterior, or posterior wall (Figure 1D). For the basilar artery (BA), locations were classified as ventral, dorsal, right, or left wall (Figure 1E).

### 2.3. Follow-Up Assessment

Patients were followed for a period exceeding one year, with a face-to-face follow-up conducted at the 12th month. The occurrence of mortality, hemorrhagic transformations, and recurrent stroke rates at 90 days and 1 year were collected.

### 2.4. Statistical Analysis

SPSS 24.0 (IBM, Armonk, NY, USA) was used for data analysis. Categorical variables were analyzed using the Chi-square test or Fisher exact test, as appropriate. The Student’s *t*-test was used to compare continuous parametric variables and the Mann–Whitney and median tests were used for nonparametric testing. A *p* value of <0.05 was considered statistically significant.

## 3. Results

Among 316 subjects who underwent HR MR-VWI, a total of 171 patients were included, including 129 patients with acute cerebral infarction or transient ischemic attack (TIA) and 42 asymptomatic patients (Figure 2). The clinical and plaque characteristics of the two groups are summarized in Table 1. Missing carotid artery results in one patient.

There was no significant difference in age between the two groups. In terms of gender, the proportion of males in the symptomatic group was 78.29%, while that in the asymptomatic group was 61.90%; the difference was statistically significant (χ^2^ = 4.454, *p* = 0.035). In the symptomatic group, 44.96% of patients smoked, and 23.81% in the asymptomatic group smoked; the difference was also statistically significant (χ^2^ = 5.918, *p* = 0.015). In addition, there was a significant difference between hypertension and diabetes. The proportion of hypertensive patients in the asymptomatic group was 47.62%, while that in the symptomatic group was 69.77% (χ^2^ = 6.773, *p* = 0.009). Patients with diabetes accounted for 33.33% of the symptomatic group and only 9.52% of the asymptomatic group (χ^2^ = 9.012, *p* = 0.003).

Imaging findings showed that the incidence of carotid atherosclerosis, family history, hyperlipidemia, and degree of enhancement between the two groups were not statistically significant, although there were some proportional differences. There were significant differences in plaque vulnerability (χ^2^ = 37.503, *p* < 0.001), incidence of IPH (χ^2^ = 5.261, *p* = 0.022), and stenosis (t = 2.675, *p* = 0.008). The symptoms group had the highest incidence of plaques located in the MCA upper wall (32.76%) and the basilar artery located in the left lateral wall (9.48%). However, the difference between the two groups did not reach statistical significance. It is worth noting that the number of cerebral infarction cases of basilar artery lesions included in this study is small.

### 3.1. Comparison of the Features of Culprit Plaque and Non-Culprit Plaque

A total of 196 intracranial arterial plaques were detected in 129 patients with ischemic stroke or TIA, of which 129 were identified as culprit plaques directly associated with ischemic stroke events and 67 were considered non-culprit plaques. As for culprit plaque, the following arterial segment involvement were intracranial internal carotid artery in 6 patients (3.4%), M1 segment of the MCA in 84 patients (65.12%), V4 segment of vertebral artery in 5 patients (3.86%), basilar artery in 32 patients (24.81%), and P1 segment of posterior cerebral artery in 1 patient (0.78%).

As shown in Table 2, culprit plaques occurred more common in the superior wall, while non-culprit plaque were more common in the lower wall and dorsal side (χ^2^ = 15.562, *p* = 0.001). The incidence of non-culprit plaques in the inferior wall was 38.18%, which was significantly higher than that of culprit plaques (18.58%). For culprit plaques in the basilar arteries, a predominance was observed on the ventral and left lateral wall (χ^2^ = 34.134 *p* = 0.000). The sample size of vertebrobasilar artery infraction included in this study was small, and further large sample studies are warranted to validate these findings.

In terms of enhancement, non-culprit plaques were more likely to show no enhancement (26.87%) and mild-to-moderate enhancement (49.25%), while 47.20% culprit plaques were significantly enhanced (χ^2^ = 23.077, *p* < 0.001). As for plaque type, non-culprit plaques were mainly classified as type VIII (37.31%), while culprit plaques were mainly classified as type IV (71.20%), with significant difference (χ^2^ = 62.038, *p* < 0.001). The analysis of IPH showed that the proportion of IPH in culprit plaques was higher (17.60%), and the difference was statistically significant (χ^2^ = 5.039, *p* = 0.025).

The stability of plaques was significantly different between the two groups. The vulnerable plaques accounted for nearly half of the culprit plaques (46.27% vs. 2.4%, χ^2^ = 57.605, *p* < 0.001). In addition, the degree of stenosis caused by culprit plaques was more severe than that caused by non-culprit plaques (40.04 ± 18.89 vs. 58.28 ± 24.44, t = −5.739, *p* < 0.001).

### 3.2. Univariate Analysis of Favorable Prognosis and Poor Prognosis

According to the one-year follow-up mRS scores, stroke patients were divided into two groups for analysis: favorable prognosis (scores 0–2) and poor prognosis (scores 3–6), as shown in Table 3. In both groups, the proportion of males exceeded that of females, yet no significant difference was found between the two groups (χ^2^ = 1.016, *p* = 0.313). The patients in the poor prognosis group were older (t = −3.709, *p* = 0.001). Box plots (Figure 3) were used to present the analysis of continuous variables between the two groups. Moreover, a higher proportion of patients in the poor prognosis group also had carotid atherosclerosis (95.45%, χ^2^ = 4.371, *p* = 0.037), suggesting that the prognosis of patients with only intracranial arteriosclerosis is superior to that of those with both intracranial and extracranial arteriosclerosis. This may also suggest that more extensive arteriosclerosis is associated with a poorer prognosis for stroke patients.

For laboratory examination, the D-dimer level as an indicator of thrombosis differed significantly between the two groups (T = −2.489, *p* = 0.022). This suggests an association between pathogenesis and stroke prognosis. Nevertheless, no significant differences were observed in HDL, LDL, HCY, leukocyte count, hs-CRP, platelet count, INR, and plasma fibrinogen levels. Baseline NIHSS scores at admission between the two groups were influencing factors for poor prognosis (2.24 ± 2.75 vs. 5.90 ± 4.84, *p* =0.003). The NIHSS score at discharge in the poor prognosis group was higher than that at admission (5.90 ± 4.84 vs. 6.81 ± 5.68), indicating that patients with progressive stroke are associated with poor prognosis and a higher disability rate.

HR MR-VWI was used to analyze the plaque characteristics between the two groups. The incidence of IPH was related to poor prognosis (χ^2^ = 3.904, *p* = 0.048), and the recognition of such plaques should be emphasized in clinical practice. The degree of stenosis was higher in the poor prognosis group, although it had not yet reached statistical significance (T = 0.937, *p* = 0.351). Previous studies have confirmed that it may be more closely related to the establishment of collateral circulation [21]. No statistical differences were observed in the location, thickness, or length of plaques either.

## 4. Discussion

ICAD is a heterogeneous disease, with multiple distinct mechanisms converging to produce downstream ischemia [22], such as rupture of vulnerable plaques, hemodynamic changes, micro embolization, and local hypo-perfusion. Identifying pathogenesis and characteristics of intracranial arterial plaque is an important basis for the treatment of stroke patients, especially in the Asian population with a high incidence of intracranial vascular disease. In this study, all accessible intracranial vascular segments in each subject were analyzed, including those without apparent lesions, and plaque characteristics were evaluated in relation to clinical prognosis through follow-up.

There were two main findings in current. First, compared with the asymptomatic group, the rates of hypertension, diabetes, and smoking were significantly higher in the symptomatic group. Plaques located in the superior wall of MCA, high stenosis, instability, and IPH were closely related to the occurrence of ischemic stroke. Second, age of onset stroke, carotid atherosclerosis, IPH, D-dimer level, and higher NIHSS score at admission and discharge were identified as factors correlated with the prognosis of ischemic stroke.

HR MR-VWI provides an in vivo method of visualizing atherosclerotic plaques, including plaques that have not led to observable stenosis [6]. It provides more useful vessel wall information in differential diagnosis and clinical management. Previous studies have proposed a strategy called “treating arteries instead of risk factors” [23], in participants with normal biochemical levels but high plaque burden and progression.

As far as we know, independent risk factors for atherosclerosis include hypertension, diabetes, hyperlipidemia, smoking, etc. There was no significant difference in the incidence of hyperlipidemia between symptomatic and asymptomatic groups in the present study. This may be attributed in part to the limited sample size and the overall high prevalence of hyperlipidemia. Additionally, hyperlipidemia subtypes were not differentiated in the analysis. Lowering blood cholesterol levels, especially LDL cholesterol, has been shown to be effective in reducing the risk of atherosclerotic disease but not triglycerides [24,25].

The study also discerned that a preponderant proportion of culprit plaques were localized on the superior wall of the MCA. Anatomically, this distribution was hypothesized to be correlated with lenticulostriate arteries preponderantly emanate from the superior wall of the MCA [26,27], and the pathogenic mechanism entailed the occlusion of branches by atherosclerotic plaques. Among the 42 BA plaques incorporated into this study, the lateral wall harbored the largest number, with 14 plaques in the left lateral wall and 7 in the right lateral wall, accounting for 50%. In the research conducted by Kim B.J et al., the majority of plaques were situated in the posterior wall, followed by the lateral wall and the anterior wall in descending order of prevalence [28]. The correlation between the location of BA plaques and ischemic stroke events remains to be validated by additional large-scale sample investigations. Yao W. et al. proposed that the evolution of atherosclerotic plaques shown by HR MRI reexamination before and after treatment can strengthen the risk stratification of patients with intracranial atherosclerosis [29].

IPH is caused by the rupture of the neovascularization [30]. Enhanced plaque is associated with abundant active inflammatory cells, neo-vessel formation, and fibrous cap thinning [31]. Higher enhancement ratio and IPH have been established as independent risk factors for long-term stroke recurrence and serve as valuable imaging markers for predicting and stratifying risk of stroke recurrence [32,33]. Consistent with the previous literature, this study found that IPH was associated with an increased risk of ischemic stroke and poor prognosis. The enhancement of culprit plaques was higher than that of non-culprit plaques, but there was no significant correlation with prognosis. The identification and treatment of plaque with IPH and significant enhancement should be considered imperative in future ischemic stroke treatment strategies, especially in asymptomatic intracranial arterial plaque.

Morphological analyses of plaques revealed no significant disparities in the thickness and length between culprit plaques and non-culprit plaques, nor between the favorable and poor prognosis groups. However, the degree of stenosis exhibited a significant variation, suggesting a role for arterial remodeling in the occurrence of ischemic stroke events. Qiao Y. et al. reported that intracranial arteries remodel in response to plaque formation, and posterior circulation arteries have a greater capacity for positive remodeling [34]. Future studies regarding the association between plaque length and thickness and clinical ischemic events should take into account the influence of arterial remodeling, which might provide more comprehensive insights into the underlying mechanisms and potential therapeutic targets.

In this cohort, all except one patient underwent extracranial vascular evaluation (carotid ultrasound, CTA, or MRA). Among these, over three-quarters were found to have concomitant carotid atherosclerosis. The proportions of cases with carotid artery plaques in the symptomatic and asymptomatic groups were 78.91% and 71.43%, respectively. The presence of both intracranial and extracranial arterial plaques was associated with a poor prognosis. A three-year follow-up study of asymptomatic participants found that the presence of carotid plaque was associated with the progression of ICAD [35]. These results underscore the importance of comprehensive vascular evaluation in clinical practice to avoid underestimating systemic atherosclerotic burden.

Notably, recent years have witnessed considerable advances in the application of artificial intelligence (AI) and machine learning to intracranial plaque imaging and stroke care [36,37,38]. ML methods have proven effective in acute stroke imaging for large vessel occlusion detection and infarct core delineation [37], and have also shown promise in improving the prediction of long-term outcomes in ischemic stroke patients [38]. A recent review of 36 AI studies in the field of stroke indicated that RapidAI demonstrates high sensitivity in detecting large vessel occlusion (87%) and acute ischemic stroke (96%), while RapidASPECTS and RapidCTA also show strong performance in stroke assessment [39]. Its implementation has enhanced the accuracy and efficiency of the radiology workflow. Furthermore, AI-based clinical decision support systems are increasingly providing clinicians with insights that extend beyond conventional tools [40], thereby contributing to more informed and personalized treatment strategies. These technologies show considerable value in diagnosing stroke types with high accuracy, enhancing the speed and precision of clinical decisions [39]. Nevertheless, the integration of AI and ML into routine clinical workflows still requires careful attention to technical, regulatory, and practical challenges to fully realize its benefits [36]. The advancement of AI and ML technologies provides strong support for subsequent research that aim to leverage these technologies to further advance this study, improve the identification rate of ICAD, and ultimately support clinical decision-making.

This study has certain limitations. Firstly, as a single-center study, it has a relatively small sample size and a short follow-up period. Only five recurrent stroke cases were recorded during the one-year follow-up, which constrained the statistical power to show the association between plaque characteristics with stroke recurrence. The underrepresentation of posterior circulation stroke also introduced potential bias in the analysis of BA plaque location. Multi-center studies with a larger sample size are needed in the future. Our team plans to continue follow-up with this cohort to further assess the value of quantitative plaque analysis in predicting recurrent stroke. Secondly, significant differences in baseline NIHSS scores between prognostic groups may have confounded the interpretation of laboratory results. The conclusion that factors such as HCY and diabetes have no significant correlation with prognosis is inconsistent with previous studies, highlighting the need for future research to incorporate more comprehensive subgroup and multivariate analyses. Finally, intracranial atherosclerosis represents a dynamic process, which may exhibit progression or regression over the passage of time. Regrettably, the follow-up imaging was not systematically performed in this study, precluding the assessment of plaque evolution and its relationship with clinical outcomes. Subsequent studies should include longitudinal imaging to better understand the natural history and therapeutic modifications of ICAD.

## 5. Conclusions

In conclusion, vulnerable plaque, IPH, location, and stenosis are associated with both the incidence and prognosis of stroke events in patients with ICAD. Enhancement grade is correlated with stroke occurrences yet demonstrates no conspicuous correlation with prognosis. Neither the length nor the thickness of plaques manifests a significant correlation with either stroke events or the prognostic outcomes. These findings possess potential implications for the clinical management of intracranial plaques and stroke prevention in patients with ICAD. With the widespread application of AI tools, further research should focus on integrating HR MR-VMI and AI technologies for primary stroke prevention and develop strategies to maximize their application in the field of stroke.

In the future, further investigation into plaque morphology, composition, and quantitative analysis through AI and ML technology is expected to facilitate the development of precise strategies for stroke prevention and treatment.

## Figures and Tables

**Figure 1 brainsci-15-01009-f001:**
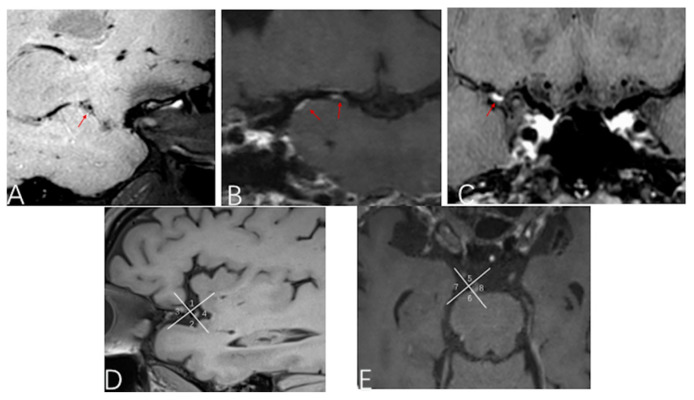
Plaque enhancement grading and location diagram. The red arrow indicates the plaques in the middle cerebral artery, which are classified into three grades based on the degree of enhancement. (**A**): No enhancement in the middle cerebral artery plaque (Grade 1); (**B**): Mild-to-moderate enhancement (Grade 2); (**C**): High enhancement (Grade 3). (**D**): Middle cerebral artery plaque location, position 1 is the superior wall, 2 is the inferior wall, 3 is the anterior wall, and 4 is the posterior wall. (**E**): Basilar artery plaque location, position 5 is the ventral wall, 6 is the dorsal wall, 7 is the right wall, and 8 is the left wall.

**Figure 2 brainsci-15-01009-f002:**
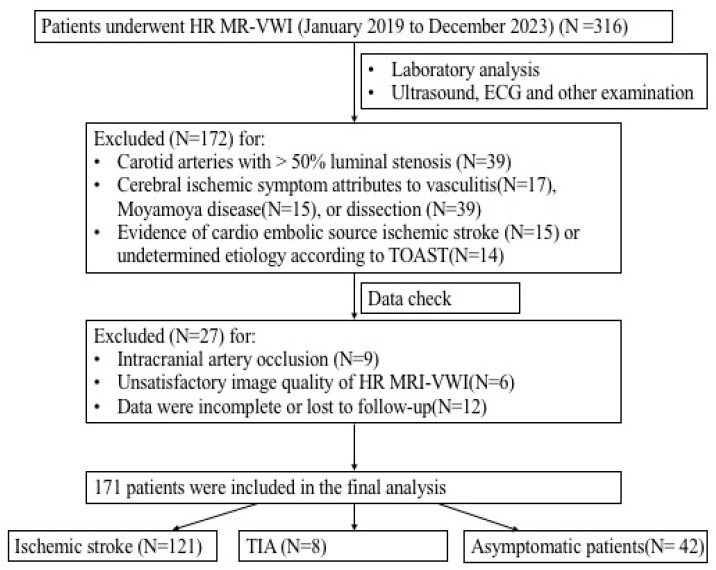
Patient selection flow chart. Among 316 subjects who underwent HR MR-VWI, a total of 171 patients were included, including 129 patients with acute cerebral infarction or transient ischemic attack (TIA) and 42 asymptomatic patients.

**Figure 3 brainsci-15-01009-f003:**
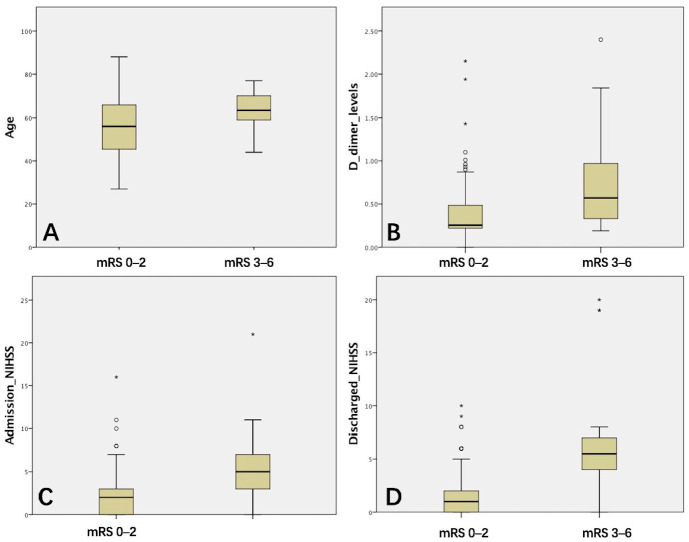
Box plots of factors influencing functional outcomes (mRS scores). The box plots illustrate the differences in age (**A**), D-dimer levels (**B**), admission NIHSS scores (**C**), and discharge NIHSS scores (**D**) between patients with favorable and poor prognosis. *: Extreme outliers.

**Table 1 brainsci-15-01009-t001:** Comparison of clinical data and plaque imaging features between symptomatic group and asymptomatic group.

Characteristics	Mean ± SD/%		Total	χ^2^/T	*p*
	Symptomatic	Asymptomatic			
Male	101 (78.29)	26 (61.90)	127 (74.27)	4.454	0.035
Age	56.99 ± 12.78	58.52 ± 10.18		−0.793	0.430
Smoking	58 (44.96)	10 (23.81)	68 (39.77)	5.918	0.015
Family history	14 (10.85)	1 (2.56)	15 (8.93)	2.53	0.112
Hypertension	90 (69.77)	20 (47.62)	110 (64.33)	6.773	0.009
Diabetes mellitus	43 (33.33)	4 (9.52)	47 (27.49)	9.012	0.003
Hyperlipemia	95 (73.64)	28 (66.67)	123 (71.93)	0.764	0.382
Carotid atherosclerosis	101 (78.91)	30 (71.43)	131 (77.06)	1.000	0.317
MCA Location					
Superior wall	38 (32.76)	14 (37.84)	52 (33.99)		
Inferior wall	22 (18.97)	16 (43.24)	38 (24.84)	5.103	0.151 *
Ventral wall	20 (17.24)	5 (13.51)	25 (16.34)		
Dorsal wall	4 (3.45)	0 (0.00)	4 (2.61)		
BA Location					
Ventral wall	9 (7.76)	1 (2.70)	10 (6.54)		
Dorsal wall	7 (6.03)	0 (0.00)	7 (4.58)		
Right lateral wall	5 (4.31)	0 (0.00)	5 (3.27)	1.436	1.00 *
Left lateral wall	11 (9.48)	1 (2.70)	12 (7.84)		
Enhancement					
Grade 1	6 (4.65)	6 (14.29)	12 (7.02)		
Grade 2	62 (48.06)	18 (42.86)	80 (46.78)	4.509	0.105
Grade 3	61 (47.29)	18 (42.86)	79 (46.20)		
IPH	23 (17.80)	2 (4.8%)	25 (14.60)	5.261	0.022
Vulnerable	126 (97.7)	27 (64.3)	153 (89.5)	37.503	<0.001
Thickness	0.22 ± 0.11	0.20 ± 0.14		1.304	0.194
Length	0.96 ± 2.92	0.73 ± 1.24		0.475	0.635
Stenosis	59.31 ± 24.30	47.75 ± 24.36		2.675	0.008

* Fisher. SD, standard deviation; MCA, middle cerebral artery; BA, basilar artery; IPH, intraplaque hemorrhage.

**Table 2 brainsci-15-01009-t002:** Image characteristics analysis of culprit plaques and non-culprit plaques.

Characteristics	Group		Total	χ^2^	*p*
	Non-Culprit	Culprit			
MCA Location				15.561	0.001 *
Superior wall	9 (16.36)	39 (34.51)	48 (28.57)		
Inferior wall	21 (38.18)	21 (18.58)	42 (25.00)		
Ventral wall	7 (12.73)	19 (16.81)	26 (15.48)		
Dorsal wall	7 (12.73)	3 (2.65)	10 (5.95)		
BA location				34.138	<0.001 *
Ventral wall	1 (1.82)	9 (7.96)	10 (5.95)		
Dorsal wall	4 (7.27)	7 (6.19)	11 (6.55)		
Right lateral wall	2 (3.64)	5 (4.42)	7 (4.17)		
Left lateral wall	4 (7.27)	10 (8.85)	14 (8.33)		
Enhancement				23.077	<0.001
1	18 (26.87)	6 (4.80)	24 (12.50)		
2	33 (49.25)	60 (48.00)	93 (48.44)		
3	16 (23.88)	59 (47.20)	75 (39.06)		
Modified AHA type				62.038	<0.001
IV	33 (49.25)	89 (71.20)	122 (63.54)		
V	0 (0.00)	1 (0.80)	1 (0.52)		
VI	3 (4.48)	32 (25.60)	35 (18.23)		
VII	6 (8.96)	1 (0.80)	7 (3.65)		
VIII	25 (37.31)	2 (1.60)	27 (14.06)		
IPH	4 (5.97)	22 (17.60)	26 (13.54)	5.039	0.025
Vulnerable	36 (53.73)	122 (97.60)	158 (82.29)	57.605	<0.001
Thickness	0.17 ± 0.09	0.26 ± 0.38		−1.926	0.056
Length	0.42 ± 0.25	0.95 ± 2.98		−1.459	0.146
Stenosis	40.04 ± 18.89	58.28 ± 24.44		−5.739	<0.001

*: Fisher. MCA, middle cerebral artery; BA, basilar artery; IPH, intraplaque hemorrhage; AHA, American Heart Association.

**Table 3 brainsci-15-01009-t003:** Analysis of favorable prognosis and poor prognosis.

Characteristics	Group (%/x ± SD)		Total	χ^2^/T	*p*
	Favorable	Poor			
Male	82 (76.64)	19 (86.36)	101 (78.29)	1.016	0.313
Age	55.63 ± 13.15	63.64 ± 8.19		−3.709	0.001 **
Smoking	51 (47.66)	7 (31.82)	58 (44.96)	1.851	0.174
Family history	14 (13.08)	0 (0.00)	14 (10.85)	3.229	0.072
Hypertension	73 (68.22)	17 (77.27)	90 (69.77)	0.708	0.400
Diabetes mellitus	33 (30.84)	10 (45.45)	43 (33.33)	1.754	0.185
Hyperlipemia	80 (74.77)	15 (68.18)	95 (73.64)	0.408	0.523
Carotid atherosclerosis	80 (75.47)	21 (95.45)	101 (78.91)	4.371	0.037 *
MCA Location					
Superior wall	31 (31.96)	7 (36.84)	38 (32.76)	1.159	0.789
Inferior wall	20 (20.62)	2 (10.53)	22 (18.97)		
Ventral wall	17 (17.53)	3 (15.79)	20 (17.24)		
Dorsal wall	4 (4.12)	0 (0.00)	4 (3.45)		
VA Location				89.086	<0.001 **
Ventral wall	6 (6.19)	3 (15.79)	9 (7.76)		
Dorsal wall	5 (5.15)	2 (10.53)	7 (6.03)		
Right lateral wall	4 (4.12)	1 (5.26)	5 (4.31)		
Left lateral wall	10 (10.31)	1 (5.26)	11 (9.48)		
Enhancement					
Grade 1	6 (5.61)	0 (0.00)	6 (4.65)	1.484	0.476
Grade 2	50 (46.73)	12 (54.55)	62 (48.06)		
Grade 3	51 (47.66)	10 (45.45)	61 (47.29)		
IPH	15 (14.29)	7 (31.82)	22 (17.32)	3.904	0.048 *
Vulnerable	102 (96.23)	22 (100.00)	124 (96.88)	0.857	0.355
Thickness	0.26 ± 0.41	0.24 ± 0.13		0.232	0.817
Length	0.92 ± 3.22	1.14 ± 1.22		−0.316	0.753
Stenosis	58.44 ± 24.64	63.82 ± 23.73		−0.937	0.351
Triglyceride	1.70 ± 0.92	1.60 ± 0.70		0.449	0.654
Total cholesterol	4.22 ± 1.17	4.54 ± 1.03		−1.101	0.273
HDL	1.08 ± 0.32	1.06 ± 0.22		0.278	0.781
LDL	2.64 ± 1.03	2.93 ± 0.94		−1.115	0.267
HCY	11.85 ± 5.16	12.02 ± 4.35		−0.128	0.898
Renal insufficiency	7 (6.5)	1 (4.5)			0.195
Leukocyte Count	7.27 ± 2.41	8.04 ± 1.29		−1.424	0.157
CRP	3.13 ± 4.19	6.34 ± 7.69		−1.563	0.138
Platelet Count	240.03 ± 61.61	247.10 ± 65.46		−0.475	0.636
INR	0.92 ± 0.07	0.94 ± 0.08		−0.907	0.366
Plasma fibrinogen levels	3.41 ± 2.41	3.51 ± 0.89		−0.183	0.855
D-dimer levels	0.40 ± 0.36	0.77 ± 0.60		−2.489	0.022 *
Admission NIHSS	2.24 ± 2.75	5.90 ± 4.84		−3.356	0.003 **
Discharged NIHSS	1.75 ± 2.19	6.81 ± 5.68		−4.013	0.001 **

* *p* < 0.05, ** *p* < 0.01. HDL, High Density Lipoprotein; LDL, Low Density Lipoprotein; HCY, Homocysteine; hs-CRP, high sensitivity C-Reactive Protein; INR, International Normalized Ratio; NIHSS, National Institute of Health Stroke Scale.

## Data Availability

The datasets supporting this study are available from the first author upon reasonable request. The data are not publicly available due to ethical reasons.

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
