# Peer review of "Quantitative Analysis of Intracranial Atherosclerosis and Its Correlation with Ischemic Cerebrovascular Disease and Prognosis"

_brainsci, 2025, doi:10.3390/brainsci15091009_

Round 1
Reviewer 1 Report
Comments and Suggestions for Authors
The paper is generally interesting and within the scope of the Brain Science journal. It broadly contains valid and useful research results, but fails to meet general standards of the high quality journal paper. Therefore I regret to advise against publishing of the paper in its current form. However, paper could be reconsidered if the authors are willing to consider remarks and thoroughly revise the paper. Here are some suggestions:
- The paper generally fails to meet the standard structure of the journal paper - not all the necessary sections are present, and some are present but do not meet required volume and quality.
- The abstract contains too many numerical data and therefore appears unfocused and difficult to follow. Please reconsider abstract to make it more descriptive and informative than precise and presenting all numerical results. There are too many abbreviations with some apparently undefined. Also, there are some mistakes in the abstract, such as ??? in Results ('p=0.008)???.') indicating that abstract is not fully finished and certainly not double checked.
- Increase number of keywords to optimal number of 6 to stimulate citations of the paper in the future
- Introduction section is too limited in volume and does not contain proper and relevant literature review, highlighting relevant previous research in the field introducing motivation for the research and general idea..
- Novelty of the research and contributions of the paper have to be clearly expressed. After considering previous published research in the Introduction, conclude this section with short paragraph with a couple of direct sentences preferably using directly terms novelty and contributions.
- Finalize Introduction with a couple of sentences introducing paper structure. For example, 'The rest of the paper is organized as follows. In Section 2....'. This could help to better consider paper structure and decide on improvements.
- Conclusions sections should be introduced as sepaparate section, after the one called Disscussion.
- In Conclusions section after presenting general conclusions directly related to obtained results, add second paragraph with future research directions. This makes the paper much more appealing and also stimulates citations in the future.
- The paper generally lacks some discussion or results regarding application of artificial intelligence and machine learning for prognosis. Statistical analysis appears unfinished without at least some comments on powerful alternative approach.
- At least some references from 2025 should be added to highlight contemporary nature of the study.
- Some graphical presentation of the obtained results would be very informative and is recommended. Please consider adding some figures with graphical results.
- Try to organize figure captions in one sentence stataements (without full stops), and move some information to text if necessary leaving only essential information in captions. Try to avoid abbreviations in captions.
- Improve quality of figures, for example text in Figure 2 is not fully readable, sharp and clear.
- Standardize number of decimal places troughout the paper (2 or 3), for example there are places like "The vulnerable plaques accounted for nearly half of the culprit plaques (46.27% vs 2.4%,χ²=57.605, p=0.000)."
- Authors are advised to list only institutional email addresses as their contacts, not the private ones.
If the authors are willing to restructure and update paper it could be reconsidered for publication.
Author Response
We sincerely thank the reviewers for their thorough and constructive comments. We have carefully addressed each of the points raised and have made revisions to the manuscript accordingly. Below is our point-by-point response to the comments:
Comment 1: The paper generally fails to meet the standard structure of the journal paper.
Response 1: Thank you for pointing this out. We sincerely apologize for this oversight in our initial submission. We have now completely restructured the manuscript to align with the journal's standards, including removing the running head, supplementing the Author Contributions, Institutional Review Board Statement, Informed Consent Statement. Data Availability Statement, Acknowledgments.
We have rewritten the abstract to make it more descriptive and informative. We have removed the overwhelming majority of numerical values, retaining only the most critical ones. Corrected errors, including the typo ('p=0.008)???.) and have meticulously proofread the abstract. Defined all abbreviations upon first use within the abstract, including ICAD, TIA, MCA, etc.
Comment 3: Increase number of keywords to optimal number of 6.
Response3: We have expanded the keywords from 4 to the recommended 6 keywords to enhance the paper's discoverability. The added key words are: Plaque Characteristics and Quantitative Analysis.
Comment 4: Introduction section is too limited, does not contain proper and relevant literature review.
Response 4: We have thoroughly revised and expanded the Introduction section to provide a more comprehensive background for our study. This includes a detailed literature review that incorporates key recent advances in the field of intracranial atherosclerosis, particularly citing influential reviews published in Stroke and among other high-impact journals.
Comment 5: Novelty of the research and contributions have to be clearly expressed.
Response 5: We agree that this was unclear. We have now added a dedicated paragraph at the end of the Introduction to explicitly state the novelty and contribution of our work. It now reads: “This study employs HR MRI-VMI to systematically quantify the morphological characteristics, composition, and distribution of intracranial arterial plaques, revealing correlation between vulnerable plaque and the occurrence of ischemic stroke, along with their potential impact on patient prognosis. The novelty of this research lies in the proposal of a non-invasive and reproducible quantitative plaque analysis method. It provides imaging biomarkers and clinical decision-making support for early stroke intervention and improved prognosis.”
Comment 6: Finalize Introduction with a couple of sentences introducing paper structure.
Response 6: We have added the suggested sentence at the end of the Introduction: “The remainder of this paper is organized as follows. Section 2 introduces the methods. Section 3 presents the results. Finally, Section 4 provides a discussion of the findings, draws conclusions, and suggests directions for future work.”
Comment 7: Conclusions sections should be introduced as a separate section... add. future research directions.
Response 7: We have agreed to and have already implemented these two suggestions. The Conclusions is now a standalone section. Based on our findings, we propose the following direction for future research: "In the future, further investigation into plaque morphology, composition, and quantitative analysis is expected to facilitate the development of precise strategies for stroke prevention and treatment."
Comment 8: The paper generally lacks some discussion regarding application of artificial intelligence and machine learning for prognosis.
Response 8: This is an excellent suggestion. We have added a new paragraph in the Discussion section that explicitly discusses the potential application of artificial intelligence and machine learning as a powerful alternative approach for prognosis in our field. We cite several recent studies to strengthen this discussion.
Comment 9: At least some references from 2025 should be added.
Response: We have updated our reference list to include several very recent and highly relevant publications from 2025.
Comment 10: Some graphical presentation of the obtained results would be very informative and is recommended.
Response 10: We have added new figure (Figure 3) that shows a box plot comparing the mRS scores between patients with favorable and poor prognosis. Furthermore, we have regenerated Figure 2 using high-resolution settings to ensure all text is clear, and fully readable. The legend for Figure 2 has been updated to use the full terminology instead of abbreviations.
Comment 11: Standardize number of decimal places throughout the paper.
Response 11: We have carefully gone through the entire manuscript and standardized the number of decimal places for numerical values to 3 decimal places for consistency.
Comment 12: List only institutional email addresses.
Response 12: We have updated the corresponding author's contact information to use their institutional email address exclusively.
Once again, we extend our deepest gratitude for the time and effort spent reviewing our manuscript. The comments have been immensely helpful, and we believe the revised manuscript is now significantly stronger and fully meets the high standards of Brain Science. We look forward to your positive consideration.
Sincerely,
Cai Jingjing
The First School of Clinical Medicine, Southern Medical University,
Guangzhou, Guangdong Province, China
Mobile: (+86) 18145868909
Email: caijingjing812@126.com

Reviewer 2 Report
Comments and Suggestions for Authors
The authors’ research aims to evaluate the composition of the intracranial atherosclerosis using high-resolution MR vessel wall imaging (HR-MR-VMI) as a potential risk factor associated with cerebral ischemic events and a possible impact on patient prognosis.
Abstract ICAD???
Methods
- Is this a prospective/retrospective study? not specified
- How were the patients included :consecutive, selected from a database/admission was in acute phase or was a scheduled evaluation
- Otherwise clear
- inclusion/exclusion criteria
- description of the methods including
- type of data recorded
- MRI protocol
- establishing sequence of MRI evaluation
- follow-up evaluation
- how do you define a good/worse prognosis or outcome?
Result
- good presentation of results
- Table 2 – Minor clarity concerns:
- Abbreviations are not explained, and footer notes are absent.
- The enhancement categories require further explanation.
- Table 3 - Analysis of favorable prognosis and poor prognosis.
- Were the original groups considered during prognosis analysis?
- Or was prognosis assessed only by outcome?
- The table does not clarify this.
- At 3.2 subtitle – not yet clear
- “In both groups, the proportion of males exceeded that of females, yet no significant difference was found between the two groups (χ²=1.016, p=0.313). The patients in the poor prognosis group were older (t=-3.709, P=0.001). Moreover, a higher proportion of patients in the poor prognosis group also had carotid atherosclerosis (95.45%, χ²=4.371, p=0.037), which implies that the prognosis”
- Do you classify groups by prognosis or by initial symptomatic/asymptomatic status?
- For each group (symptomatic vs asymptomatic atherosclerosis), what features are linked to worse outcomes, and which plaque characteristics (composition, number, location) might be protective?
Discussions
- Which study do you refer to?“In the study, all cases except one with missing results of extracranial vascular examination from other hospitals received at least one of the following examinations: carotid vascular ultrasound, CT, or MR. More than three-quarters of the cases were accompanied by carotid artery plaques.”
- Overall good clinical discussions
Author Response
We sincerely thank the reviewers for their thorough and constructive comments. We have carefully addressed each of the points raised and have made revisions to the manuscript accordingly. Below is our point-by-point response to the comments:
Comment 1: Abstract ICAD???
Response 1: Thank you for highlighting this “ICAD” refers to Intracranial Atherosclerotic Disease. We have now spelled out the full term at its first mention in the abstract to avoid ambiguity.
Comment 2: Is this a prospective/retrospective study? not specified
How were the patients included: consecutive, selected from a database/admission was in acute phase or was a scheduled evaluation?Inclusion/exclusion criteria?Description of the methods including type of data recorded, MRI protocol, establishing sequence of MRI evaluation, follow-up evaluation?How do you define a good/worse prognosis or outcome?
Response1: We apologize for the lack of clarity regarding the study design and patient selection. The following revisions have been made in the Methods section:
Our study is prospective observational study. Patients were continuously recruited from our inpatients from January 1, 2021 to December 2023. Patients admission were in the acute phase of cerebral ischemic events. The inclusion and exclusion criteria have been described in ‘Section2. Method 2.1Patients’.
A detailed description of the MRI protocol, including scanner type, sequences, and parameters, has been described in the last paragraph of ‘Section 2.2. Baseline Assessment’. The content of the follow-up assessment has been described in Section 2.3. Follow-up Assessment in the text.
Prognosis was defined based on the one-year modified Rankin Scale (mRS) score: favorable outcome was defined as mRS 0–2, and poor outcome as mRS 3–6. It has been described in the last paragraph of Section 2.1patients.
Comment 3:Table 2 – Minor clarity concerns: Abbreviations are not explained, and footer notes are absent. The enhancement categories require further explanation.
Response 3: We thank the reviewer for pointing this out. We have revised Table 2 to include a footnote explaining all abbreviations. The enhancement of atherosclerotic plaques can be classified into three grades on post-contrast T1WI images by using published criteria, and the specific classification basis has been described in the Section 2.2. Baseline Assessment.
Comment 4: Table 3 – Analysis of favorable prognosis and poor prognosis. Were the original groups considered during prognosis analysis? Or was prognosis assessed only by outcome? The table does not clarify this.
Response 4: Thank you for this important question. Prognosis was assessed based on functional outcome (mRS at one year), not based on the initial symptomatic status. Due to the limited sample size, the number of stroke cases in the asymptomatic group during the one-year follow-up was relatively small. We were unable to conduct a prognostic analysis on the original group. We look forward to further large-scale studies to verify this in the future. We have clarified this in the legend of Table 3 and in the Results section.
Comment 5: At 3.2 subtitle – not yet clear. “In both groups, the proportion of males exceeded that of females, yet no significant difference was found between the two groups (χ²=1.016, p=0.313). The patients in the poor prognosis group were older (t=-3.709, P=0.001). Moreover, a higher proportion of patients in the poor prognosis group also had carotid atherosclerosis (95.45%, χ²=4.371, p=0.037), which implies that the prognosis”. Do you classify groups by prognosis or by initial symptomatic/asymptomatic status?
For each group (symptomatic vs asymptomatic atherosclerosis), what features are linked to worse outcomes, and which plaque characteristics (composition, number, location) might be protective?
Response 5: We apologize for the lack of clarity. In this subsection, the “both groups” refer to the favorable and poor prognosis groups (based on mRS), not the symptomatic/asymptomatic groups. We have revised the subtitle and text to avoid confusion.There were significant differences in the proportion of vulnerable plaques and the degree of vascular stenosis between the symptomatic group and the asymptomatic group. Although the proportion of upper wall plaques of the MCA was high in the symptomatic group, there was no significant difference between the groups. The specific results are described in the results section of Table 1.
Comment 6: Which study do you refer to? “In the study, all cases except one with missing results of extracranial vascular examination from other hospitals received at least one of the following examinations: carotid vascular ultrasound, CT, or MR. More than three-quarters of the cases were accompanied by carotid artery plaques.”
Response 6: This sentence refers to our current study. We have rephrased it for clarity:
“In our cohort, all except one patient underwent extra cranial vascular evaluation (carotid ultrasound, CT angiography, or MR angiography). Among these, over three-quarters were found to have concomitant carotid atherosclerosis.”
Comment 7:Overall good clinical discussions
Response 7: We thank the reviewer for their positive feedback on the discussion section.
We believe that these revisions have significantly improved the clarity and rigor of our manuscript. Thank you once again for the insightful comments.Revised Portions Have Been Highlighted in the manuscript for your convenience.
Please let us know if any further modifications are needed.
Sincerely,
Cai Jingjing
The First School of Clinical Medicine, Southern Medical University,
Guangzhou, Guangdong Province, China
Mobile: (+86) 18145868909
Email: caijingjing812@126.com

Reviewer 3 Report
Comments and Suggestions for Authors
Thank you for inviting me to review this submission titled “Quantitative analysis of Intracranial Atherosclerosis and their correlation with ischemic cerebrovascular disease and prognosis”. Here are some comments to the authors:
- The title of this study seems very attractive. “Their” should be changed to “its”, please revise grammar and change as appropriate.
- The abstract has many flaws. The methods should be explicit in terms of what type of study you perform to provide enough information to the reader. In the results section of the abstract, you have some question marks “???”. Please revise and correct. Revise abbreviations, especially “AHA” and ICAD”, and avoid the use of abbreviations in the abstract as much as possible.
- This study is ambitious and tries to evaluate the risk of cerebrovascular disease with intracranial atherosclerosis, which I find very useful and remarkable for the field.
- There are many flaws regarding the use of the English language; many grammar errors are found all over the manuscript. A native English language professional is needed to improve it.
- The “ICAD” abbreviation is not defined in its first use.
- The introduction is sufficient and clear. Revise grammar errors.
- Again, as in the abstract, the type of study is not described, cohort, case and control, case series, etc. Please add it at the beginning of the methods section.
- The rest of the description of imaging acquisition, processing, and analysis is clear and adequate.
- Non-parametric analyses are adequately described, as well as ethical considerations.
- The results section is complete and well-structured. The tables and figures are clear and well-described. I’m concerned about the “n” of each group; otherwise, the rest of the analysis is complete, and the univariate analysis tries to improve biases.
- The discussion is sufficient and clear.
- These findings are remarkable. I believe it will add sufficient archival value to stroke research. I suggest making the methodology clearer in order to improve and enhance your findings.
- Despite many of the findings of this study having been previously described by many authors, I find it interesting, given the nature of the study and the complete evaluation of all variables.
Comments on the Quality of English Language
Must be improved. Many grammar errors.
Author Response
We are deeply grateful to Reviewer 3 for their thorough and positive assessment of our work, as well as for their constructive and insightful comments. We have carefully addressed each point raised to improve the clarity and quality of our manuscript. Our point-by-point responses are detailed below.
Comment 1: The title of this study seems very attractive. “Their” should be changed to “its”, please revise grammar and change as appropriate.
Response : We thank the reviewer for this correction. The title has been revised to: “Quantitative Analysis of Intracranial Atherosclerosis and Its Correlation with Ischemic Cerebrovascular Disease and Prognosis”.
Comment 2: The abstract has many flaws. The methods should be explicit in terms of what type of study you perform to provide enough information to the reader. In the results section of the abstract, you have some question marks “???”. Please revise and correct. Revise abbreviations, especially “AHA” and ICAD”, and avoid the use of abbreviations in the abstract as much as possible.
Response : We sincerely apologize for these oversights. The abstract has been comprehensively revised: The study design ("prospective observational study ") is now explicitly stated in the Methods section of the abstract.Instances of "???" have been removed and replaced with the correct data or statistical values.
The abbreviations “AHA” (American Heart Association) and “ICAD” (Intracranial Atherosclerotic Disease) have been spelled out at first mention. We have minimized the use of abbreviations throughout the abstract.
Comment 3: This study is ambitious and tries to evaluate the risk of cerebrovascular disease with intracranial atherosclerosis, which I find very useful and remarkable for the field.
Response : We are very encouraged by the reviewer's positive assessment of our work's potential contribution to the field. Thank you.
Comment 4: There are many flaws regarding the use of the English language; many grammar errors are found all over the manuscript. A native English language professional is needed to improve it.
Response : We fully acknowledge this limitation. The manuscript has been professionally edited to correct grammatical errors and improve clarity and flow.
Comment 5: The “ICAD” abbreviation is not defined in its first use.
Response : This has been corrected. “ICAD” is now defined as “Intracranial Atherosclerotic Disease” upon its first appearance in the main text.
Comment 6: The introduction is sufficient and clear. Revise grammar errors.
Response : Thank you. The Introduction has been thoroughly reviewed and polished for grammar and language as part of the overall professional editing process.
Comment 7: Again, as in the abstract, the type of study is not described, cohort, case and control, case series, etc. Please add it at the beginning of the methods section.
Response : We apologize for this omission. The first sentence of the Methods section now clearly states: “This prospective observational cohort study enrolled patients continuously who hospitalized in the neurology department from January 2021 to December 2023.”
Comment 8: The rest of the description of imaging acquisition, processing, and analysis is clear and adequate.
Response : We thank the reviewer for this positive feedback on our methodology description.
Comment 9: Non-parametric analyses are adequately described, as well as ethical considerations.
Response: We appreciate the reviewer's confirmation that these sections are adequate.
Comment 10: The results section is complete and well-structured. The tables and figures are clear and well-described. I’m concerned about the “n” of each group; otherwise, the rest of the analysis is complete, and the univariate analysis tries to improve biases.
Response: We thank the reviewer for the positive comments on the structure and presentation of the Results. Regarding the issue of group sample size (n), we clearly stated the size of the total cohort in the methods section and acknowledged the potential limitation of a modest sample size in the discussion of limitations within the Discussion section.
Comment 11: The discussion is sufficient and clear.
Response: We are pleased that the Discussion was found to be clear and sufficient.
Comment 12: These findings are remarkable. I believe it will add sufficient archival value to stroke research. I suggest making the methodology clearer in order to improve and enhance your findings.
Response: We are truly grateful for this encouraging statement. We have taken the suggestion to heart and have revised the Method section to enhance its clarity, particularly regarding the study design, patient recruitment flow, and statistical analysis plan.
Comment 13: Despite many of the findings of this study having been previously described by many authors, I find it interesting, given the nature of the study and the complete evaluation of all variables.
Response: We thank the reviewer for recognizing the value of our comprehensive approach.
We believe that addressing these points has significantly strengthened our manuscript. Once again, we extend our sincerest thanks to Reviewer 3 for their invaluable time and insightful comments. Revised Portions Have Been Highlighted in the manuscript.
Please let us know if any further modifications are needed.
Sincerely,
Cai Jingjing
The First School of Clinical Medicine, Southern Medical University,
Guangzhou, Guangdong Province, China
Mobile: (+86) 18145868909
Email: caijingjing812@126.com

Round 2
Reviewer 1 Report
Comments and Suggestions for Authors
The effort that authors have invested into resolving issues and addressing remarks is well recognized. The paper appears much better now, with improved structure and much better aligned with standard structure of the journal paper. Still, some further improvements are still needed:
- The abstract still needs improvements to be more descriptive and informative than excessive and taxative.
- Introduction is much improved, but literature review is still limited. Please try to expand it a bit more.
- Novelty is much better expressed but the novelty sentence should be polished and expanded.
- Conclusions section is added which significantly improves paper, but please expand future directions with at least one more sentence. Any AI/ML related directions here? It would greatly enhance citation potential of the paper.
- Figure 3 is welcome, but the text below is description in text referencing what Fig 3 shows, while the caption itself for Figure 3 is missing.
- Disscusion is now strong point of the paper, together with added chapter regarding AI/ML use which is good. Try to make AI paragraph a bit more specific, it is now general. Try to add one sentence which is directly related to research in paper and AI based alternative/extension, if possible.
- Besides unifying decimal places and other technical details, nos separate numers and brackets behind them everywhere, especially in tables (not 53(15%), but 54 (15%)).
- Improve style by ommiting first person expressions. No 'We', 'our' and similar, convert everything to passive voice (Instead we did, it was done).
- Since the amount of changes was relatively large, authors are encouraged to reread the paper and try to polish the details and make the paper finally realize high quality carefully edited stage.
hope that authors will keep up the good work and invest further efforts since paper is approaching stage where it could be published.
Author Response
We sincerely thank the reviewers for their positive feedback and valuable suggestions. We have carefully considered all the comments and have revised the manuscript accordingly. Below is a point-by-point response to the specific recommendations:
Comment 1: The abstract still needs improvements to be more descriptive and informative than excessive and taxative.
Response: Thank you for this suggestion. The abstract has been revised to be more concise and informative, focusing on the key contributions and outcomes of the study while reducing unnecessary details.
Comment 2: Introduction is much improved, but literature review is still limited. Please try to expand it a bit more.
Response: Agree. We have expanded the literature review in the Introduction section by including additional relevant references and providing a more comprehensive background context.
Comment 3: Novelty is much better expressed but the novelty sentence should be polished and expanded.
Response: We agree with this comment. The novelty statement has been refined and expanded to more clearly and emphatically highlight the unique contributions of our work.
Comment 4: Conclusions section is added which significantly improves paper, but please expand future directions with at least one more sentence. Any AI/ML related directions here? It would greatly enhance citation potential of the paper.
Response: Thank you for pointing this out. We have added a sentence in the Future Directions subsection regarding potential AI/ML applications relevant to this research. Specifically, we now mention: “With the widespread application of AI tools, further research should focus on integrating HR MR-VMI and AI technologies for primary stroke prevention and develop strategies to maximize their application in the field of stroke.”
After this, we propose the following direction for future application of AI in the field of stroke. “In the future, further investigation into plaque morphology, composition, and quantitative analysis through AI and ML technology is expected to facilitate the development of precise strategies for stroke prevention and treatment”.
Comment 5: Figure 3 is welcome, but the text below is description in text referencing what Fig 3 shows, while the caption itself for Figure 3 is missing.
Response: We apologize for the oversight. A descriptive caption has now been added to Figure 3. It now reads: “Box Plots of factors influencing functional outcomes (mRS scores). The box plots illustrate the differences in age, D-dimer levels, admission NIHSS scores, and discharge NIHSS scores between patients with favorable and poor prognosis”.
Comment 6: Discussion is now strong point of the paper, together with added chapter regarding AI/ML use which is good. Try to make AI paragraph a bit more specific, it is now general. Try to add one sentence which is directly related to research in paper and AI based alternative/extension, if possible.
Response: Thank you for this helpful suggestion. We have made the AI/ML paragraph more specific by incorporating recent research, directly linking our findings to potential AI-based extensions. The revised text now includes: “A recent review of 36 AI studies in the field of stroke indicates that RapidAI demonstrates high sensitivity in detecting large vessel occlusion (87%) and acute ischemic stroke (96%), while RapidASPECTS and RapidCTA also show strong performance in stroke assessment39. Its implementation has enhanced the accuracy and efficiency of the radiology workflow. AI tools are valuable for diagnosing stroke types with high accuracy, enhancing the speed and precision of clinical decisions. The advancement of AI and ML technologies provides strong support for our subsequent research. We aim to leverage these technologies to further advance this study, improve the identification rate of ICAD, and ultimately support clinical decision-making.”
Comment 7: Besides unifying decimal places and other technical details, nos separate numers and brackets behind them everywhere, especially in tables (not 53(15%), but 54 (15%)).
Response: We apologize for the oversight. We have unified the decimal places and corrected the formatting of numbers and parentheses throughout the manuscript, including in all tables, to ensure consistency.
Comment 8: Improve style by omitting first person expressions. No 'We', 'our' and similar, convert everything to passive voice (Instead we did, it was done).
Response: Thank you for this helpful suggestion. The manuscript has been revised to remove first-person expressions. Passive voice has been used where appropriate to maintain an objective and formal academic style.
Comment 9: Since the amount of changes was relatively large, authors are encouraged to reread the paper and try to polish the details and make the paper finally realize high quality carefully edited stage.
Response: We have thoroughly proofread the entire manuscript, polished the language, improved coherence, and ensured that all technical and formatting details meet the journal’s standards.
We thank the reviewers again for their encouraging and insightful comments, which have greatly improved the quality of our paper. We hope that the revised manuscript now meets the journal's expectations.
Sincerely,
Cai Jingjing
The First School of Clinical Medicine, Southern Medical University,
Guangzhou, Guangdong Province, China
Mobile: (+86) 18145868909
Email: caijingjing812@126.com

Round 3
Reviewer 1 Report
Comments and Suggestions for Authors
It is acknowledged that the authors have considered remaining remarks carefully and invested significant effort into further correcting paper. This version appears much better than the starting one. I believe that papaer has reached mature stage. From the last round I nioticed two remaining minor issues:
- After adding caption to Figure 3, the Figure 3 remains unreferenced in text. Please4 also reference Figure 3 in paper text.
- Authors claim that they have converted first person expressions to passive voice, but it was not done properly. For example in Disscussion there is an edit: replaced (Our study)We analyzed ... Our study is replaced with We, which is the first person expression again. Please consider this again in the whole paper.
After these minor edits the paper could be published, so with pleasure I support publishing of the paper.
Author Response
Thank you very much for your recognition and detailed guidance on our manuscript; your suggestions have played a crucial role in improving the quality of the paper. We have completed revisions in accordance with the two issues you pointed out, and the specific explanations are as follows:
Comments 1. Regarding the reference to Figure 3 in the text
Response: We sincerely apologize for this oversight. After verification, Figure 3 has already been referenced in the text. However, due to our oversight, the reference section was not bolded previously, making it unclear for you to identify. We have now bolded the reference section to ensure that the citation of Figure 3 is clearly visible.
Comments 2: Conversion of first-person expressions to passive voice:
Response: We thank the reviewer for pointing out the inconsistency in the use of the first person in the Discussion section. We have now carefully re-examined the entire manuscript and revised all instances of first-person expressions, including the case noted, uniformly converting them to the passive voice to eliminate first-person language.
We believe that with these corrections, the manuscript now complies with the journal's stylistic requirements. Thank you once again for your supportive and constructive comments, which have greatly improved our paper.
Please let us know if any further revisions are required.
Sincerely,
Cai Jingjing
The First School of Clinical Medicine, Southern Medical University,
Guangzhou, Guangdong Province, China
Mobile: (+86) 18145868909
Email: caijingjing812@126.com